# MCP: Learning Composable Hierarchical Control with Multiplicative Compositional Policies

**Xue Bin Peng, Michael Chang, Grace Zhang, Pieter Abbeel, Sergey Levine**
Department of Electrical Engineering and Computer Science
University of California, Berkeley
{xbpeng, mbchang, grace.zhang}@berkeley.edu
pabbeel@cs.berkeley.edu
svlevine@eecs.berkeley.edu

## Abstract

Humans are able to perform a myriad of sophisticated tasks by drawing upon skills acquired through prior experience. For autonomous agents to have this capability, they must be able to extract reusable skills from past experience that can be recombined in new ways for subsequent tasks. Furthermore, when controlling complex high-dimensional morphologies, such as humanoid bodies, tasks often require coordination of multiple skills simultaneously. Learning discrete primitives for every combination of skills quickly becomes prohibitive. Composable primitives that can be recombined to create a large variety of behaviors can be more suitable for modeling this combinatorial explosion. In this work, we propose multiplicative compositional policies (MCP), a method for learning reusable motor skills that can be composed to produce a range of complex behaviors. Our method factorizes an agent's skills into a collection of primitives, where multiple primitives can be activated simultaneously via multiplicative composition. This flexibility allows the primitives to be transferred and recombined to elicit new behaviors as necessary for novel tasks. We demonstrate that MCP is able to extract composable skills for highly complex simulated characters from pre-training tasks, such as motion imitation, and then reuse these skills to solve challenging continuous control tasks, such as dribbling a soccer ball to a goal, and picking up an object and transporting it to a target location. (Video[1])

## 1 Introduction

Reinforcement learning is commonly applied to solve tasks from scratch. While *tabula rasa* learning can achieve state-of-the-art performance on a broad range of tasks [4, 13, 27, 29, 39], this approach can incur significant drawbacks in terms of sample efficiency and limits the complexity of skills that an agent can acquire. The ability to transfer and re-purpose skills learned from prior experiences to new domains is a hallmark of intelligent agents. Transferable skills can enable agents to solve tasks that would otherwise be prohibitively challenging to learn from scratch, by leveraging prior experiences to provide structured exploration and more effective representations. However, learning versatile and reusable skills that can be applied to a diverse set of tasks remains a challenging problem, particularly when controlling systems with large numbers of degrees-of-freedom.

In this work, we propose multiplicative compositional policies (MCP), a method for learning reusable motor primitives that can be composed to produce a continuous spectrum of skills. Once learned, the primitives can be transferred to new tasks and combined to yield different behaviors as necessary in the target domain. Standard hierarchical models [10, 41] often activate only a single primitive at each timestep, which can limit the diversity of behaviors that can be produced by the agent. MCP

composes primitives through a multiplicative model that enables multiple primitives to be activated at a given timestep, thereby providing the agent a more flexible range of skills. Our method can therefore be viewed as providing a means of composing skills in space, while standard hierarchical models compose skills in time by temporally sequencing the set of available skills. MCP can also be interpreted as a variant of latent space models, where the latent encoding specifies a particular composition of a discrete set of primitives.

The primary contribution of our work is a method for learning and composing transferable skills using multiplicative compositional policies. By pre-training the primitives to imitate a corpus of different motion clips, our method learns a set of primitives that can be composed to produce a flexible range of behaviors. While conceptually simple, MCP is able to solve a suite of challenging mobile manipulation tasks with complex simulated characters, significantly outperforming prior methods as task complexity grows. Our analysis shows that the primitives discover specializations that are reminiscent of previous manually-designed control structures, and produce coherent exploration strategies that are vital for high-dimensional long-horizon tasks. In our experiments, MCP substantially outperforms prior methods for skill transfer, with our method being the only approach that learns a successful policy on the most challenging task in our benchmark.

## 2   Preliminaries

We consider a multi-task RL framework for transfer learning, consisting of a set of pre-training tasks and transfer tasks. An agent is trained from scratch on the pre-training tasks, but it may then apply any skills learned during pre-training to the subsequent transfer tasks. The objective then is to leverage the pre-training tasks to acquire a set of reusable skills that enables the agent to be more effective at the later transfer tasks. Each task is represented by a state space $s_t \in \mathcal{S}$, an action space $a_t \in \mathcal{A}$, a dynamics model $s_{t+1} \sim p(s_{t+1}|s_t, a_t)$, a goal space $g \in \mathcal{G}$, a goals distribution $g \sim p(g)$, and a reward function $r_t = r(s_t, a_t, g)$. The goal specifies task specific features, such as a motion clip to imitate, or the target location an object should be placed. All tasks share a common state space, action space, and dynamics model. However, the goal space, goal distribution, and reward function may differ between pre-training and transfer tasks. For each task, the agent's objective is to learn an optimal policy $\pi^*$ that maximizes its expected return $J(\pi) = \mathbb{E}_{g \sim p(g), \tau \sim p_\pi(\tau|g)} \left[ \sum_{t=0}^{T} \gamma^t r_t \right]$ over the distribution of goals from the task, where $p_\pi(\tau|g) = p(s_0) \prod_{t=0}^{T-1} p(s_{t+1}|s_t, a_t)\pi(a_t|s_t, g)$ denotes the distribution over trajectories $\tau$ induced by the policy $\pi$ for a given goal $g$. $T$ represents the time horizon, and $\gamma \in [0, 1]$ is the discount factor. Successful transfer cannot be expected for unrelated tasks. Therefore, we consider the setting where the pre-training tasks encourage the agent to learn relevant skills for the subsequent transfer tasks, but may not necessarily cover the full range of skills required to be effective at the transfer tasks.

Hierarchical policies are a common model for reusing and composing previously learned skills. One approach for constructing a hierarchical policy is by using a mixture-of-experts model [15, 19, 28, 31, 42], where the composite policy's action distribution $\pi(a|s, g)$ is represented by a weighted sum of distributions from a set of primitives $\pi_i(a|s, g)$ (i.e. low-level policies). A gating function determines the weights $w_i(s, g)$ that specify the probability of activating each primitive for a given $s$ and $g$,

$$\pi(a|s, g) = \sum_{i=1}^{k} w_i(s, g)\pi_i(a|s, g), \quad \sum_{i=1}^{k} w_i(s, g) = 1, \quad w_i(s, g) \geq 0. \quad (1)$$

Here, $k$ denotes the number of primitives. We will refer to this method of composing primitives as an *additive model*. To sample from the composite policy, a primitive $\pi_i$ is first selected according to $w$, then an action is sampled from the primitive's distribution. Therefore, a limitation of the additive model is that only one primitive can be active at a particular timestep. While complex behaviors can be produced by sequencing the various primitives in time, the action taken at each timestep remains restricted to the behavior prescribed by a single primitive. Selecting from a discrete set of primitive skills can be effective for simple systems with a small number of actuated degrees-of-freedom, where an agent is only required to perform a small number of subtasks at the same time. But as the complexity of the system grows, an agent might need to perform more and more subtasks *simultaneously*. For example, a person can walk, speak, and carry an object all at the same time. Furthermore, these subtasks can be combined in any number of ways to produce a staggering array of diverse behaviors. This combinatorial explosion can be prohibitively challenging to model with policies that activate only one primitive at a time.

# 3 Multiplicative Compositional Policies

In this work, we propose multiplicative compositional policies (MCP), a method for composing primitives that addresses this combinatorial explosion by explicitly factoring the agent's behavior – not with respect to time, but with respect to the action space. Our model enables the agent to activate multiple primitives simultaneously, with each primitive specializing in different behaviors that can be composed to produce a continuous spectrum of skills. Our probabilistic formulation accomplishes this by treating each primitive as a distribution over actions, and the composite policy is obtained by a multiplicative composition of these distributions,

$$\pi(a|s,g) = \frac{1}{Z(s,g)} \prod_{i=1}^{k} \pi_i(a|s,g)^{w_i(s,g)}, \quad w_i(s,g) \geq 0. \tag{2}$$

Unlike an additive model, which activates only a single primitive per timestep, the *multiplicative model* allows multiple primitives to be activated simultaneously. The gating function specifies the weights $w_i(s,g)$ that determine the influence of each primitive on the composite action distribution, with a larger weight corresponding to a larger influence. The weights need not be normalized, but in the following experiments, the weights will be bounded $w_i(s,g) \in [0,1]$. $Z(s,g)$ is the partition function that ensures the composite distribution is normalized. While the additive model directly samples actions from the selected primitive's distribution, the multiplicative model first combines the primitives, and then samples actions from the resulting distribution.

## 3.1 Gaussian Primitives

Gaussian policies are a staple for continuous control tasks, and modeling multiplicative primitives using Gaussian policies provides a particularly convenient form for the composite policy. Each primitive $\pi_i(a|s,g) = \mathcal{N}(\mu_i(s,g), \Sigma_i(s,g))$ will be modeled by a Gaussian with mean $\mu_i(s,g)$ and diagonal covariance matrix $\Sigma_i(s,g) = \mathrm{diag}\left(\sigma_i^1(s,g), \sigma_i^2(s,g), ..., \sigma_i^{|\mathcal{A}|}\right)$, where $\sigma_i^j(s,g)$ denotes the variance of the $j$th action parameter from primitive $i$, and $|\mathcal{A}|$ represents the dimensionality of the action space. A multiplicative composition of Gaussian primitives yields yet another Gaussian policy $\pi(a|s,g) = \mathcal{N}(\mu(s,g), \Sigma(s,g))$. Since the primitives model each action parameter with an independent Gaussian, the action parameters of the composite policy $\pi$ will also assume the form of independent Gaussians with component-wise mean $\mu^j(s,g)$ and variance $\sigma^j(s,g)$,

$$\mu^j(s,g) = \frac{1}{\sum_{l=1}^{k} \frac{w_l(s,g)}{\sigma_l^j(s,g)}} \sum_{i=1}^{k} \frac{w_i(s,g)}{\sigma_i^j(s,g)} \mu_i^j(s,g), \qquad \sigma^j(s,g) = \left(\sum_{i=1}^{k} \frac{w_i(s,g)}{\sigma_i^j(s,g)}\right)^{-1}. \tag{3}$$

Note that while $w_i(s,g)$ determines a primitive's overall influence on the composite distribution, each primitive can also independently adjust its influence per action parameter through $\sigma_i^j(s,g)$. Once the parameters of the composite distribution have been determined, $\pi$ can be treated as a regular Gaussian policy, and trained end-to-end using standard automatic differentiation tools.

## 3.2 Pre-Training and Transfer

The primitives are learned through a set of pre-training tasks. The same set of primitives is responsible for solving all pre-training tasks, which results in a collection of primitives that captures the range of behaviors needed for the set of tasks. Note, the primitives are not manually assigned to particularly tasks. Instead, the primitives are trained jointly in an end-to-end fashion and the specializations emerge automatically from the learning process. Algorithm 1 illustrates the overall training process.

---
**Algorithm 1** MCP Pre-Training and Transfer
---
1: Pre-training:
2: $\pi_i \leftarrow$ random parameters for $i = 1, ..., k$
3: $w \leftarrow$ random parameters
4: $\pi_{1:k}^*, w^* = \underset{\pi_{1:k}, w}{\arg\max} \; J_{pre}(\pi_{1:k}, w)$
5: Transfer:
6: $\omega \leftarrow$ random parameters
7: $\omega^* = \underset{\omega}{\arg\max} \; J_{tra}(\pi_{1:k}^*, \omega)$
---

$J_{pre}(\pi_{1:k}, w)$ denotes the objective for the pre-training tasks for a given set of primitives $\pi_{1:k}$ and gating function $w$, and $J_{tra}(\pi_{1:k}, \omega)$ denotes the objective for the transfer tasks. When transferring primitives to a new task, the parameters of the primitives are kept fixed, while a new policy is trained

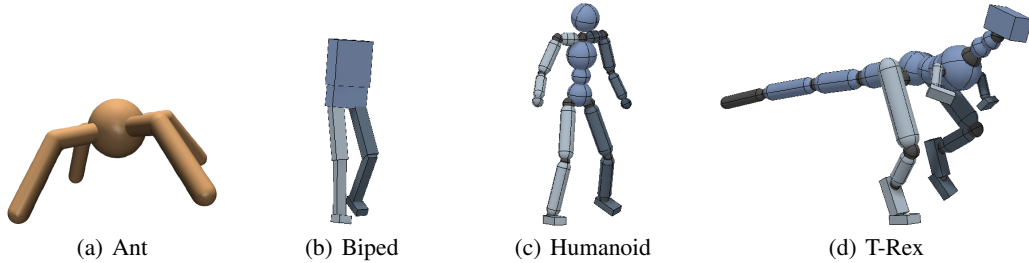

|     |     |     |     |
| --- | --- | --- | --- |
| (a) Ant | (b) Biped | (c) Humanoid | (d) T-Rex |

Figure 1: Our method is evaluated on complex 3D characters with different morphologies and large numbers of degrees-of-freedom.

to specify weights for composing the primitives. Therefore, the primitives can be viewed as a set of nonlinear basis functions that defines a new action space for use in subsequent tasks. During pre-training, in order to force the primitives to specialize in distinct skills, we use an asymmetric model, where only the gating function $w_i(s, g)$ observes the goal $g$, and the primitives have access only to the state $s$,

$$\pi(a|s, g) = \frac{1}{Z(s, g)} \prod_{i=1}^{k} \pi_i(a|s)^{w_i(s,g)}, \quad \pi_i(a|s) = \mathcal{N}\left(\mu_i(s), \Sigma_i(s)\right). \tag{4}$$

This asymmetric model prevents the degeneracy of a single primitive becoming responsible for all goals, and instead encourages the primitives to learn distinct skills that can then be composed by the gating function as needed for a given goal. Furthermore, since the primitives depend only on the state, they can be conveniently transferred to new tasks that share similar state spaces but may have different goal spaces. When transferring the primitives to new tasks, the parameters of the primitives $\pi_i(a|s)$ are kept fixed to prevent catastrophic forgetting, and a new gating function $\omega(w|s, g)$ is trained to specify the weights $w = (w_1, w_2, ...)$ for composing the primitives.

## 4 Related Work

Learning reusable representations that are transferable across multiple tasks has a long history in machine learning [1, 5, 30, 33, 43]. Finetuning remains a popular transfer learning technique when using neural network, where a model is first trained on a source domain, and then the learned features are reused in a target domain by finetuning via backpropagation [8, 18]. One of the drawbacks of this procedure is catastrophic forgetting, as backpropagation is prone to destroying previously learned features before the model is able to utilize them in the target domain [21, 34, 35].

**Hierarchical Policies:** A popular method for combining and reusing skills is by constructing hierarchical policies, where a collection of low-level controllers, which we will refer to as primitives, are integrated together with the aid of a gating function that selects a suitable primitive for a given scenario [2, 15, 41]. A common approach for building hierarchical policies is to first train a collection of primitives through a set of pre-training tasks, which encourages each primitive to specialize in distinct skills [6, 12, 24, 25, 31]. Once trained, the primitives can be integrated into a hierarchical policy and transferred to new tasks. End-to-end methods have also been proposed for training hierarchical policies [2, 7, 23, 44]. However, since standard hierarchical policies only activate one primitive at a time, it is not as amenable for composition or interpolation of multiple primitives in order to produce new skills.

**Latent Space Models:** Our work falls under a class of methods that we will refer to broadly as latent space models. These methods specify controls through a latent representation that is then mapped to the controls (i.e. actions) of the underlying system [22]. Similar to hierarachical models, a latent representation can first be learned using a set of pre-training tasks, before transferring to downstream tasks [14, 17]. But unlike a standard hierarchical model, which activates a single primitive at a time, continuous latent variables can be used to enable more flexible interpolation of skills in the latent space. Various diversity-promoting pre-training techniques have been proposed for encouraging the latent space to model semantically distinct behaviors [9, 11, 16]. Demonstrations can also be incorporated during pre-training to acquire more complex skills [26]. In this work, we present

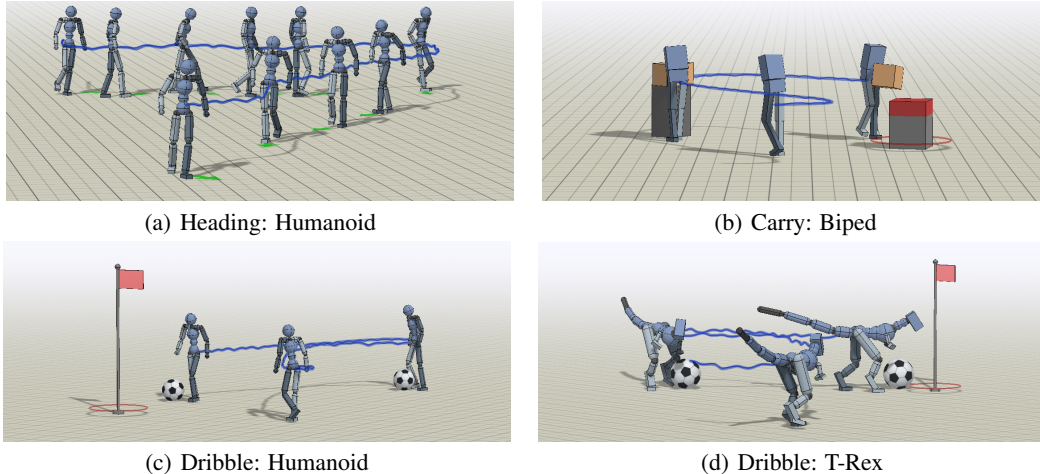

|   |   |
|:-:|:-:|
| (a) Heading: Humanoid | (b) Carry: Biped |
| (c) Dribble: Humanoid | (d) Dribble: T-Rex |

Figure 2: The transfer tasks pose a challenging combination of locomotion and object manipulation, such as carrying an object to a target location and dribbling a ball to a goal, which requires coordination of multiple body parts and temporally extended behaviors.

a method for modeling latent skill representations as a composition of multiplicative primitives. We show that the additional structure introduced by the primitives enables our agents to tackle complex continuous control tasks, achieving competitive performance when compared to previous models, and significantly outperforming prior methods as task complexity grows.

## 5   Experiments

We evaluate the effectiveness of our method on controlling complex simulated characters, with large numbers of degrees-of-freedom (DoFs), to perform challenging long-horizon tasks. The tasks vary from simple locomotion tasks to difficult mobile manipulation tasks. The characters include a simple 14 DoF ant, a 23 DoF biped, a more complex 34 DoF humanoid, and a 55 DoF T-Rex. Illustrations of the characters are shown in Figure 1, and examples of transfer tasks are shown in Figure 2. Our experiments aim to study MCP's performance on complex temporally extended tasks, and examine the behaviors learned by the primitives. We also evaluate our method comparatively to determine the value of multiplicative primitives as compared to more standard additive mixture models, as well as to prior methods based on options and latent space embeddings. Behaviors learned by the policies are best seen in the supplementary video[1].

### 5.1   Experimental Setup

**Pre-Training Tasks:**   The pre-training tasks in our experiments consist of motion imitation tasks, where the objective is for the character to mimic a corpus of different reference motions. Each reference motion specifies a sequence of target states $\{\hat{s}_0, \hat{s}_1, ..., \hat{s}_T\}$ that the character should track at each timestep. We use a motion imitation approach following Peng et al. [32]. But instead of training separate policies for each motion, a single policy, composed of multiple primitives, is trained to imitate a variety of motion clips. To imitate multiple motions, the goal $g_t = (\hat{s}_{t+1}, \hat{s}_{t+2})$ provides the policy with target states for the next two timesteps. A reference motion is selected randomly at the start of each episode. To encourage the primitives to learn to transition between different skills, the reference motion is also switched randomly to another motion within each episode. The corpus of motion clips is comprised of different walking and turning motions.

**Transfer Tasks:**   We evaluate our method on a set of challenging continuous control tasks, involving locomotion and object manipulation using the various characters. Detailed descriptions of each task are available in the supplementary material.

*Heading:* First we consider a simple heading task, where the objective is for the character to move along a target heading direction $\hat{\theta}_t$. The goal $g_t = (\cos(\hat{\theta}_t), -\sin(\hat{\theta}_t))$ encodes the heading as a unit vector along the horizontal plane. The target heading varies randomly over the course of an episode.

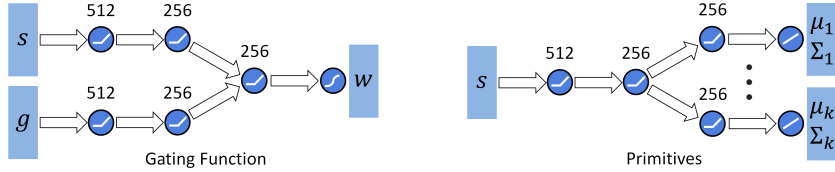

Figure 3: Schematic illustrations of the MCP architecture. The gating function receives both $s$ and $g$ as inputs, which are first encoded by separate networks, with 512 and 256 units. The resulting features are concatenated and processed with a layer of 256 units, followed by a sigmoid output layer to produce the weights $w(s, g)$. The primitives receive only $s$ as input, which is first processed by a common network, with 512 and 256 units, before branching into separate layers of 256 units for each primitive, followed by a linear output layer that produces $\mu_i(s)$ and $\Sigma_i(s)$ for each primitive. ReLU activation is used for all hidden units.

| Environment | Scratch | Finetune | Hierarchical | Option-Critic | MOE | Latent Space | MCP (Ours) |
|---|---|---|---|---|---|---|---|
| Heading: Biped | $0.927 \pm 0.032$ | $0.970 \pm 0.002$ | $0.834 \pm 0.001$ | $0.952 \pm 0.012$ | $0.918 \pm 0.002$ | $0.970 \pm 0.001$ | $\mathbf{0.976 \pm 0.002}$ |
| Carry: Biped | $0.027 \pm 0.035$ | $0.324 \pm 0.014$ | $0.001 \pm 0.002$ | $0.346 \pm 0.011$ | $0.013 \pm 0.013$ | $0.456 \pm 0.031$ | $\mathbf{0.575 \pm 0.032}$ |
| Dribble: Biped | $0.072 \pm 0.012$ | $0.651 \pm 0.025$ | $0.546 \pm 0.024$ | $0.046 \pm 0.008$ | $0.073 \pm 0.021$ | $0.768 \pm 0.012$ | $\mathbf{0.782 \pm 0.008}$ |
| Dribble: Humanoid | $0.076 \pm 0.024$ | $0.598 \pm 0.030$ | $0.198 \pm 0.002$ | $0.058 \pm 0.007$ | $0.043 \pm 0.021$ | $0.751 \pm 0.006$ | $\mathbf{0.805 \pm 0.006}$ |
| Dribble: T-Rex | $0.065 \pm 0.032$ | $0.074 \pm 0.011$ | $-$ | $0.098 \pm 0.013$ | $0.070 \pm 0.017$ | $0.115 \pm 0.013$ | $\mathbf{0.781 \pm 0.021}$ |
| Holdout: Ant | $\mathbf{0.951 \pm 0.093}$ | $0.885 \pm 0.062$ | $-$ | | | $0.745 \pm 0.060$ | $0.812 \pm 0.030$ |

Table 1: Performance statistics of different models on transfer tasks. Additional experiments are available in the supplementary material. MCP outperforms other methods on a suite of challenging tasks with complex simulated characters.

*Carry:* To evaluate our method's performance on long horizon tasks, we consider a mobile manipulation task, where the objective is to move a box from a source location to a target location. The task can be decomposed into a sequence of subtasks, where the character must first pickup the box from the source location, before carrying it to the target location, and placing it on the table. The source and target are placed randomly each episode. Depending on the initial configuration, the task may require thousands of timesteps to complete. The goal $g_t = (x_{tar}, q_{tar}, x_{src}, q_{src}, x_b, q_b, v_b, \omega_b)$ encodes the target's position $x_{tar}$ and orientation $q_{tar}$ represented as a quaternion, the source's position $x_{src}$ and orientation $q_{src}$, and box's position $x_b$, orientation $q_b$, linear velocity $v_b$, and angular velocity $\omega_b$.

*Dribble:* This task poses a challenging combination of locomotion and object manipulation, where the objective is to move a soccer ball to a target location. Since the policy does not have direct control over the ball, it must rely on complex contact dynamics in order to manipulate the movement of the ball while also maintaining balance. The location of the ball and target are randomly initialized each episode. The goal $g_t = (x_{tar}, x_b, q_b, v_b, \omega_b)$ encodes the target location $x_{tar}$, and ball's position $x_b$, orientation $q_b$, linear velocity $v_b$, and angular velocity $\omega_b$.

**Model Representation:** All experiments use a similar network architecture for the policy, as illustrated in Figure 3. Each policy is composed of $k = 8$ primitives. The gating function and primitives are modeled by separate networks that output $w(s, g)$, $\mu_{i:k}(s)$, and $\Sigma_{i:k}(s)$, which are then composed according to Equation 2 to produce the composite policy. The state describes the configuration of the character's body, with features consisting of the relative positions of each link with respect to the root, their rotations represented by quaternions, and their linear and angular velocities. Actions from the policy specify target rotations for PD controllers positioned at each joint. Target rotations for 3D spherical joints are parameterized using exponential maps. The policies operate at 30Hz and are trained using proximal policy optimization (PPO) [37].

## 5.2 Comparisons

We compare MCP to a number of prior methods, including a baseline model trained from scratch for each transfer task, and a model first pre-trained to imitate a reference motion before being finetuned on the transfer tasks. To evaluate the effects of being able to activate and compose multiple primitives simultaneously, we compare MCP to models that activate only one primitive at a time, including a hierarchical model that sequences a set of pre-trained skills [24, 25], an option-critic model [2], and a mixture-of-experts model (MOE) analogous to Equation 1. Finally, we also include comparisons to a continuous latent space model with an architecture similar to Hausman et al. [16] and Merel et al. [26]. All models, except for the scratch model, are pre-trained with motion imitation [32]. Detailed descriptions of each method can be found in the supplementary material. Figure 4 illustrates learning

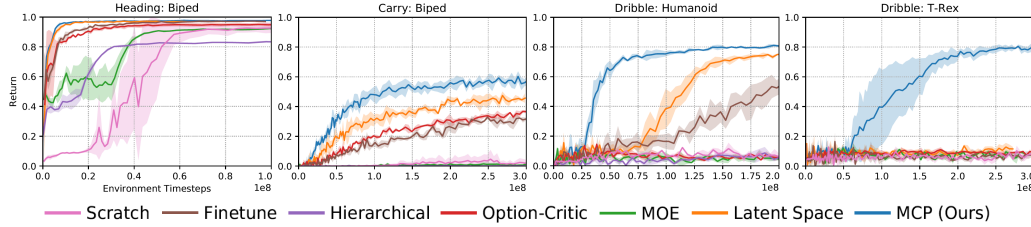

Figure 4: Learning curves of the various models when applied to transfer tasks. MCP substantially improves learning speed and performance on challenging tasks (e.g. carry and dribble), and is the only method that succeeds on the most difficult task (Dribble: T-Rex).

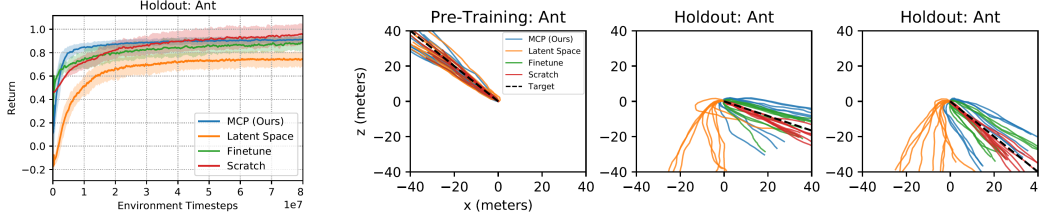

Figure 5: **Left:** Learning curves on holdout tasks in the Ant environment. **Right:** Trajectories produced by models with target directions from pre-training, and target directions from the holdout set after training on transfer tasks. The latent space model is prone to overfitting to the pre-training tasks, and can struggle to adapt to the holdout tasks.

curves for the various methods on the transfer tasks and Table 1 summarizes their performance. Each environment is denoted by "Task: Character". Performance is recorded as the average normalized return across approximately 100 episodes, with 0 being the minimum possible return per episode and 1 being the maximum. Three models initialized with different random seeds are trained for each environment and method.

Our experiments show that MCP performs well across the suite of tasks. For simple tasks such as heading, all models show similar performance. But as task complexity increases, MCP exhibits significant improvements to learning speed and asymptotic performance. Training from scratch is effective for the simple heading task, but is unable to solve the more challenging carry and dribble tasks. Finetuning proved to be a strong baseline, but struggles with the more complex morphologies. With higher dimensional action spaces, independent action noise is less likely to produce useful behaviors. Models that activate only a single primitive at a time, such as the hierarchical model, option-critic model, and MOE model, tend to converge to lower asymptotic performance due to their limited expressivity. MOE is analogous to MCP where only a single primitive is active at a time. Despite using a similar number of primitives as MCP, being able to activate only one primitive per timestep limits the variety of behaviors that can be produced by MOE. This suggests that the flexibility of MCP to compose multiple primitives is vital for more sophisticated tasks. The latent space model shows strong performance on most tasks. But when applied to characters with more complex morphologies, such as the humanoid and T-Rex, MCP consistently outperforms the latent space model, with MCP being the only model that solves the dribbling task with the T-Rex.

We hypothesize that the performance difference between MCP and the latent space model may be due to the process through which a latent code $w$ is mapped to an action for the underlying system. With the latent space model, the pre-trained policy $\pi(a|s, w)$ acts as a decoder that maps $w$ to a distribution over actions. We have observed that this decoder has a tendency to overfit to the pre-training behaviors, and can therefore limit the variety of behaviors that can be deployed on the transfer tasks. In the case of MCP, if $\sigma_i^j$ is the same across all primitives, then we can roughly view $w$ as specifying a convex combination of the primitive means $\mu_{i:k}$. Therefore, $\mu_{1:k}$ forms a convex hull in the original action space, and the transfer policy $\omega(w|s, g)$ can select any action within this set. As such, MCP may provide the transfer policy with a more flexible range of skills than the latent space model. To test this hypothesis, we evaluate the different models on transferring to out-of-distribution tasks using a simple setup. The environment is a variant of the standard Gym Ant environment [4], where the agent's objective is to run along a target direction $\hat{\theta}$. During pre-training, the policies are trained with directions $\hat{\theta} \in [0, 3/2\pi]$. During transfer, the directions are sampled from a holdout set $\hat{\theta} \in [3/2\pi, 2\pi]$. Figure 5 illustrates the learning curves on the transfer task, along with the trajectories

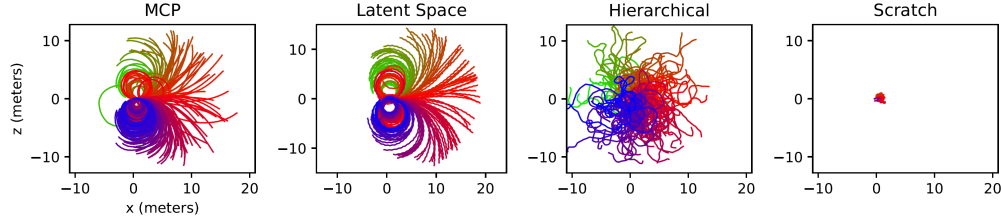

Figure 6: Trajectories of the humanoid's root along the horizontal plane visualizing the exploration behaviors of different models. MCP and other models that are pre-trained with motion imitation produce more structured exploration behaviors.

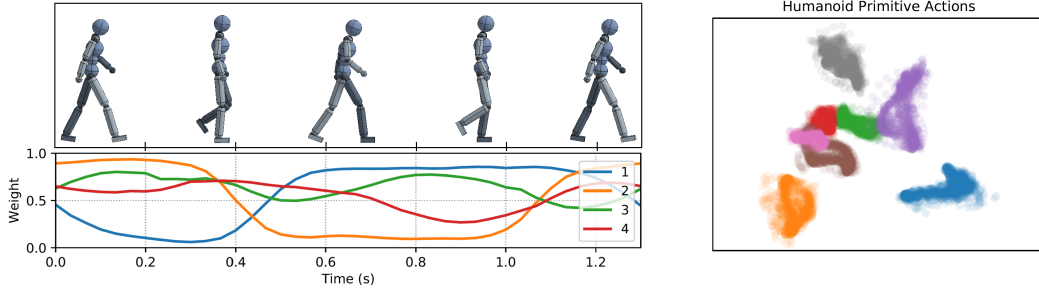

Figure 7: **Left:** Weights for primitives over the course of a walk cycle. Primitives develop distinct specializations, with some primitives becoming most active during the left stance phase, and others during right stance. **Right:** PCA embedding of actions from each primitive exhibits distinct clusters.

produced by the models when commanded to follow different target directions from the pre-training and transfer tasks. Indeed we see that the latent space model is prone to overfitting to the directions from pre-training, and struggles to adapt to the holdout directions. MCP provides the transfer policy sufficient flexibility to adapt quickly to the transfer tasks. The scratch and finetune models also perform well on the transfer tasks, since they operate directly on the underlying action space.

### 5.3 Exploration Behaviors

To analyze the exploration behaviors produced by the primitives, we visualize the trajectories obtained by random combinations of the primitives, where the weights are sampled from a Gaussian and held fixed over the course of a trajectory. Figure 6 illustrates the trajectories of the humanoid's root produced by various models. Similar to MCP, the trajectories from the latent space model are also produced by sampling $w$ from a Gaussian. The trajectories from the hierarchical model are generated by randomly sequencing the set of primitives. The model trained from scratch simply applies Gaussian noise to the actions, which leads to a fall after only few timesteps. Models that are pre-trained with motion imitation produce more structured behaviors that travel in different directions.

### 5.4 Primitive Specializations

To analyze the specializations of the primitives, we record the weight of each primitive over the course of a walk cycle. Figure 7 illustrates the weights during pre-training, when the humanoid is trained to imitate walking motions. The activations of the primitives show a strong correlation to the phase of a walk cycle, with primitive 1 becoming most active during left stance and becoming less active during right stance, while primitive 2 exhibits precisely the opposite behavior. The primitives appear to have developed a decomposition of a walking gait that is commonly incorporated into the design of locomotion controllers [45]. Furthermore, these specializations consistently appear across multiple training runs. Next, we visualize the actions proposed by each primitive. Figure 7 shows a PCA embedding of the mean action from each primitive. The actions from each primitive form distinct clusters, which suggests that the primitives are indeed specializing in different behaviors.

## 6 Conclusion

We presented multiplicative compositional policies (MCP), a method for learning and composing skills using multiplicative primitives. Despite its simplicity, our method is able to learn sophisticated behaviors that can be transferred to solve challenging continuous control tasks with complex simulated agents. Once trained, the primitives form a new action space that enables more structured exploration

and provides the agent with the flexibility to combine the primitives in novel ways in order to elicit new behaviors for a task. Our experiments show that MCP can be effective for long horizon tasks and outperforms prior methods as task complexity grows. While MCP provides a form of spatial abstraction, we believe that incorporating temporal abstractions is an important direction. During pre-training, some care is required to select an expressive corpus of reference motions. In future work, we wish to investigate methods for recovering sophisticated primitive skills without this supervision.

## Acknowledgements

We would like to thank AWS, Google, and NVIDIA for providing computational resources. This research was funded by an NSERC Postgraduate Scholarship, a Berkeley Fellowship for Graduate Study, an NSF Graduate Research Fellowship, Berkeley DeepDrive, Honda, ARL DCIST CRA W911NF-17-2-0181, Intel, and Sony Interactive Entertainment America.

## Footnotes

[1]Supplementary video: xbpeng.github.io/projects/MCP/

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
