[Supplementary Material]

## Supplementary Material

## A    Gaussian Composition Derivation

In this section, we review a proof that the weighted product of $k$ univariate Gaussian primitives $\pi_i(x) = \mathcal{N}(\mu_i, \sigma_i)$, with mean $\mu_i$, variance $\sigma_i$, and weight $w_i$, results in a composite Gaussian distribution $\pi(x)$ with mean $\mu$ and variance $\sigma$ given by:

$$\pi(x) = \frac{1}{Z} \prod_{i=1}^{k} \pi_i(x)^{w_i} = \mathcal{N}(\mu, \sigma) \tag{5}$$

$$\mu = \frac{1}{\sum_{i=1}^{k} \sum_{i=1}^{k} \frac{w_i}{\sigma_i} \mu_i}, \qquad \sigma = \left( \sum_{i=1}^{k} \frac{w_i}{\sigma_i} \right)^{-1}, \tag{6}$$

where $Z$ is the normalilzation constant that ensures the composite distribution is normalized. We start by writing out the expression of the product of Gaussian primitives

$$\pi(x) = \frac{1}{Z} \prod_i \pi_i(x)^{w_i}$$

$$= \frac{1}{Z} \prod_i \exp\left( -\frac{w_i}{2\sigma_i} (x - \mu_i)^2 \right) \tag{7}$$

$$= \frac{1}{Z} \exp\left( -\sum_i \frac{w_i}{2\sigma_i} (x - \mu_i)^2 \right).$$

Let $\sigma_{-i} = \prod_{j \neq i} \sigma_j$,

$$= \frac{1}{Z} \exp\left( \frac{-\sum_i w_i \sigma_{-i} (x - \mu_i)^2}{2 \prod_i \sigma_i} \right)$$

$$= \frac{1}{Z} \exp\left( \frac{-\sum_i w_i \sigma_{-i} (x^2 - 2x\mu_i + \mu_i^2)}{2 \prod_i \sigma_i} \right) \tag{8}$$

$$= \frac{1}{Z} \exp\left( \frac{-1}{2 \prod_i \sigma_i} \left( \left( \sum_i w_i \sigma_{-i} \right) x^2 - 2x \left( \sum_i w_i \sigma_{-i} \mu_i \right) + \sum_i w_i \sigma_{-i} \mu_i^2 \right) \right).$$

Multiplying the exponent by $\frac{\sum_i w_i \sigma_{-i}}{\sum_i w_i \sigma_{-i}}$ we get,

$$= \frac{1}{Z} \exp\left( \frac{-\sum_i w_i \sigma_{-i}}{2 \prod_i \sigma_i} \left( x^2 - 2x \frac{\sum_i w_i \sigma_{-i} \mu_i}{\sum_i w_i \sigma_{-i}} + \frac{\sum_i w_i \sigma_{-i} \mu_i^2}{\sum_i w_i \sigma_{-i}} \right) \right)$$

$$= \frac{1}{Z} \exp\left( -\frac{1}{2} \left( \sum_i \frac{w_i}{\sigma_i} \right) \left( x^2 - 2x \frac{\sum_i w_i \sigma_{-i} \mu_i}{\sum_i w_i \sigma_{-i}} + \frac{\sum_i w_i \sigma_{-i} \mu_i^2}{\sum_i w_i \sigma_{-i}} \right) \right). \tag{9}$$

Next, we complete the squares

$$= \frac{1}{Z} \exp\left( -\frac{1}{2} \left( \sum_i \frac{w_i}{\sigma_i} \right) \left( x^2 - 2x \frac{\sum_i w_i \sigma_{-i} \mu_i}{\sum_i w_i \sigma_{-i}} + \left( \frac{\sum_i w_i \sigma_{-i} \mu_i}{\sum_i w_i \sigma_{-i}} \right)^2 - \left( \frac{\sum_i w_i \sigma_{-i} \mu_i}{\sum_i w_i \sigma_{-i}} \right)^2 \right.\right.$$

$$\left.\left. + \frac{\sum_i w_i \sigma_{-i} \mu_i^2}{\sum_i w_i \sigma_{-i}} \right) \right)$$

$$= \frac{1}{Z} \exp\left( -\frac{1}{2} \left( \sum_i \frac{w_i}{\sigma_i} \right) \left( \left( x - \frac{\sum_i w_i \sigma_{-i} \mu_i}{\sum_i w_i \sigma_{-i}} \right)^2 - \left( \frac{\sum_i w_i \sigma_{-i} \mu_i}{\sum_i w_i \sigma_{-i}} \right)^2 + \frac{\sum_i w_i \sigma_{-i} \mu_i^2}{\sum_i w_i \sigma_{-i}} \right) \right)$$

$$= \frac{1}{Z} \exp\left( -\frac{1}{2} \left( \sum_i \frac{w_i}{\sigma_i} \right) \left( \left( x - \frac{1}{\sum_i \frac{w_i}{\sigma_i}} \sum_i \frac{w_i}{\sigma_i} \mu_i \right)^2 - \left( \frac{\sum_i w_i \sigma_{-i} \mu_i}{\sum_i w_i \sigma_{-i}} \right)^2 + \frac{\sum_i w_i \sigma_{-i} \mu_i^2}{\sum_i w_i \sigma_{-i}} \right) \right).$$

$$\tag{10}$$

Finally, since $-\left(\frac{\sum_i w_i \sigma_{-i} \mu_i}{\sum_i w_i \sigma_{-i}}\right)^2 + \frac{\sum_i w_i \sigma_{-i} \mu_i^2}{\sum_i w_i \sigma_{-i}}$ is independent of $x$, it can be subsumed into the normalization constant $Z$, resulting in the desired expression for the composition distribution

$$\pi(x) = \frac{1}{Z}\exp\left(-\frac{1}{2}\left(\sum_i \frac{w_i}{\sigma_i}\right)\left(x - \frac{1}{\sum_i \frac{w_i}{\sigma_i}}\sum_i \frac{w_i}{\sigma_i}\mu_i\right)^2\right)$$

$$= \mathcal{N}\left(\frac{1}{\sum_i \frac{w_i}{\sigma_i}}\sum_i \frac{w_i}{\sigma_i}\mu_i, \quad \left(\sum_i \frac{w_i}{\sigma_i}\right)^{-1}\right). \tag{11}$$

## B  Additional Experiments

A comprehensive set of learning curves for all transfer tasks are available in Figure 9 and Table 2 summarizes the performance of the final policies. Note that the hierarchical model selects a new primitive at the start of each walk cycle, approximately 30 environment timesteps, and as such operates at a lower frequency than the other models. Instead of recording the number of policy steps, we record the number of environment timestep. This corresponds to the amount of physical interactions that the agent requires to learn a policy, which is often the bottleneck for simulated and real world domains.

To analyze the effects of the number of primitives used, we trained MCP models with $k = 4, 8, 16, 32$ primitives. Figure 4 illustrates the learning curves with varying numbers of primitives. We do not observe a noticeable performance difference between 4 and 8 primitives. But as the number of primitives increases, learning efficiency appears to decrease. In the case of 32 primitives, the dimensionality of $w$ is larger than the dimensionality of the original action space for the humanoid (28D), which diminishes some of the benefits of the dimensionality reduction provided by the primitives.

When transferring primitives to new tasks, we train a new gating function for composing the primitives for the new task while keeping the parameters of the primitives fixed. To test the effects of this design decision, we compare the performance of policies on transfer tasks where only the gating function is trained for the new task (Train Gating), and policies where both the gating function and primitives are trained jointly on the transfer tasks (Train Gating + Prims). Figure 8 compares learning curves for fixing or finetuning the primitives on various transfer tasks. Overall, the performance of fixing vs finetuning the primitives lead to similar performance on most tasks. Fixing the primitives appears to lead to more significant improvements on harder tasks, such as those with the humanoid. Since no reference motions are used during training on the transfer tasks, finetuning the primitives tend to lead to more unnatural behaviors.

Figure 8: Learning curves comparing policies where only the gating function is trained for the transfer tasks, while keeping the parameters of the primitives fixed, and policies where both the gating function and primitives are trained for the new tasks. Overall, these different design decisions show similar performance on most tasks.

## C  Reference Motions

During pre-training, the primitives are trained by imitating a corpus of reference motions. The biped and humanoid share the same set of reference motions, consisting of mocap clips of walking and turning motions collected from a publicly available database [38]. In total, 230 seconds of motion

Figure 9: Learning curves of the various models when applied to transfer tasks. MCP improves learning speed and performance on challenging tasks (e.g. carry and dribble), and is the only method that succeeds on the most difficult task (Dribble: T-Rex).

| Environment | Scratch | Finetune | Hierarchical | Option-Critic | MOE | Latent Space | MCP (Ours) |
|---|---|---|---|---|---|---|---|
| Heading: Biped | $0.927 \pm 0.032$ | $0.970 \pm 0.002$ | $0.834 \pm 0.001$ | $0.952 \pm 0.012$ | $0.918 \pm 0.002$ | $0.970 \pm 0.001$ | $\mathbf{0.976 \pm 0.002}$ |
| Heading: Humanoid | $0.965 \pm 0.010$ | $\mathbf{0.975 \pm 0.008}$ | $0.681 \pm 0.006$ | $0.958 \pm 0.001$ | $0.857 \pm 0.018$ | $0.969 \pm 0.002$ | $0.970 \pm 0.003$ |
| Heading: T-Rex | $0.840 \pm 0.003$ | $\mathbf{0.953 \pm 0.004}$ | — | $0.830 \pm 0.004$ | $0.672 \pm 0.011$ | $0.686 \pm 0.003$ | $0.932 \pm 0.007$ |
| Carry: Biped | $0.027 \pm 0.035$ | $0.324 \pm 0.014$ | $0.001 \pm 0.002$ | $0.346 \pm 0.011$ | $0.013 \pm 0.013$ | $0.456 \pm 0.031$ | $\mathbf{0.575 \pm 0.032}$ |
| Dribble: Biped | $0.072 \pm 0.012$ | $0.651 \pm 0.025$ | $0.546 \pm 0.024$ | $0.046 \pm 0.008$ | $0.073 \pm 0.021$ | $0.768 \pm 0.012$ | $\mathbf{0.782 \pm 0.008}$ |
| Dribble: Humanoid | $0.076 \pm 0.024$ | $0.598 \pm 0.030$ | $0.198 \pm 0.002$ | $0.058 \pm 0.007$ | $0.043 \pm 0.021$ | $0.751 \pm 0.006$ | $\mathbf{0.805 \pm 0.006}$ |
| Dribble: T-Rex | $0.065 \pm 0.032$ | $0.074 \pm 0.011$ | — | $0.098 \pm 0.013$ | $0.070 \pm 0.017$ | $0.115 \pm 0.013$ | $\mathbf{0.781 \pm 0.021}$ |
| Holdout: Ant | $\mathbf{0.951 \pm 0.093}$ | $0.885 \pm 0.062$ | — | — | — | $0.745 \pm 0.060$ | $0.812 \pm 0.030$ |

Table 2: Performance statistics of different models on transfer tasks.

data is used to train the biped and humanoid. To retarget the humanoid reference motions to the biped, we simply removed extraneous joints in the upper body (e.g. arms and head). The reference motions for the T-Rex consist of artist generated keyframe animations. Due to the cost of manually authored animations, the T-Rex is trained with substantially less motion data than the other characters. In total, 11 seconds of motion data is used to train the T-Rex. The T-Rex motions include 1 forward walk, 2 left turns, and 2 right turns. Despite having access to only a small corpus of reference motions, MCP is nonetheless able to learn a flexible set of primitives that enables the complex T-Rex character to perform challenging tasks.

## D  Transfer Tasks

**Heading:**  First we consider a simple heading task, where the objective is for the character to move in a target heading direction $\hat{\theta}_t$. The heading is changed every timestep by applying a random perturbation $\hat{\theta}_t = \hat{\theta}_{t-1} + \nabla\theta_t$ sampled from a uniform distribution $\nabla\theta_t \sim \text{Uniform}(-0.15\text{rad}, 0.15\text{rad})$.

| Property | Biped | Humanoid | T-Rex |
|----------|-------|----------|-------|
| Links | 12 | 13 | 20 |
| Total Mass (kg) | 42 | 45 | 54.5 |
| Height (m) | 1.34 | 1.62 | 1.66 |
| Degrees-of-Freedom | 23 | 34 | 55 |
| State Features | 105 | 196 | 261 |
| Action Parameters | 17 | 28 | 49 |

Table 3: Properties of the characters.

Table 4: Learning curves of MCP with different numbers of primitives $k$.

The goal $g_t = (\cos(\hat{\theta}_t), -\sin(\hat{\theta}_t))$ encodes the heading as a unit vector along the horizontal plane. The reward $r_t$ encourages the character to follow the target heading, and is computed according to

$$r_t = \exp\left(-4 \ (\hat{u} \cdot v_{com} - \hat{v})^2\right).$$

Here, $(\cdot)$ denotes the dot product, $v_{com}$ represents the character's center-of-mass (COM) velocity along the horizontal plane, $\hat{v} = 1m/s$ represents the target speed that the character should travel in along the target direction $\hat{u} = (\cos(\hat{\theta}_t), -\sin(\hat{\theta}_t))$.

**Carry:** To evaluate our method's performance on long horizon tasks, we consider a mobile manipulation task, where the goal is for the character to move a box from a source location to a target location. The task can be decomposed into a sequence of subtasks, where the character must first pickup the object from the source location, before carrying it to the target location and placing it on the table. To enable the character to carry the box, when the character makes contact with the box at the source location with a specific link (e.g. torso), a virtual joint is created that attaches the box to the character. Once the box has been placed at the target location, the joint is detached. The box has a mass of 5kg and is initialized to a random source location at a distance of $[0m, 10m]$ from the character. The target is initialized to a distance $[0m, 10m]$ from the source. The goal $g_t = (x_{tar}, q_{tar}, x_{src}, q_{src}, x_b, q_b, v_b, \omega_b)$ encodes the target table's position $x_{tar}$ and orientation $q_{tar}$ as represented as a quaternion, the source table's position $x_{src}$ and orientation $q_{src}$, and box's position $x_b$, orientation $q_b$, linear velocity $v_b$, and angular velocity $\omega_b$. The reward function consists of terms that encourage the character to move towards the box, as well as to move the box towards the target,

$$r_t = w^{cv} r_t^{cv} + w^{cp} r_t^{cp} + w^{bv} r_t^{bv} + w^{bp} r_t^{bp},$$

$r_t^{cv}$ encourages the character to move towards the box, while $r_t^{cp}$ encourages the character to stay near the box,

$$r_t^{cv} = \exp\left(-1.5 \min\left(0, u_b \cdot v_{com} - \hat{v}\right)^2\right)$$

$$r_t^{cp} = \exp\left(-0.25 \left\|x_{com} - x_b\right\|^2\right).$$

$u_b$ represents the unit vector pointing in the direction of the box with respect to the character's COM, $v_{com}$ is the COM velocity of the character, $\hat{v} = 1m/s$ is the target speed, $x_{com}$ is the COM position, and $x_b$ is the box's position. All quantities are expressed along the horizontal plane. Similarly, $r_t^{bv}$ and $r_t^{bp}$ encourages the character to move the box towards the target,

$$r_t^{bv} = \exp\left(-1 \min\left(0, u_{tar} \cdot v_b - \hat{v}\right)^2\right)$$

$$r_t^{bp} = \exp\left(-0.5 \left\|x_b - x_{tar}\right\|^2\right).$$

$u_{tar}$ represents the unit vector pointing in the direction of the target with respect to the box, $v_b$ is the velocity of the box, and $x_{tar}$ is the target location. The weights for the reward terms are specified according to $(w^{cv}, w^{cp}, w^{bv}, w^{bp}) = (0.1, 0.2, 0.3, 0.4)$.

**Dribble:** This task poses a challenging combination of locomotion and object manipulation, where the goal is for the character to move a soccer ball to a target location. Since the policy does not have direct control over the ball, it must rely on complex contact dynamics in order to manipulate the movement of the ball while also maintaining balance. The ball is randomly initialized at a distance of $[0\text{m}, 10\text{m}]$ from the character, and the target is initialized to a distance of $[0\text{m}, 10\text{m}]$ from the ball. The goal $g_t = (x_{tar}, x_b, q_b, v_b, \omega_b)$ encodes the target location $x_{tar}$, and ball's position $x_b$, orientation $q_b$, linear velocity $v_b$, and angular velocity $\omega_b$. The reward function for this task follows a similar structure as the reward for the carry task, consisting of terms that encourage the character to move towards the ball, as well as to move the ball towards the target,

$$r_t = w^{cv} r_t^{cv} + w^{cp} r_t^{cp} + w^{bv} r_t^{bv} + w^{bp} r_t^{bp},$$

$r_t^{cv}$ encourages the character to move towards the ball, while $r_t^{cp}$ encourages the character to stay near the ball,

$$r_t^{cv} = \exp\left(-1.5 \min\left(0, u_b \cdot v_{com} - \hat{v}\right)^2\right)$$

$$r_t^{cp} = \exp\left(-0.5 \left\|x_{com} - x_b\right\|^2\right).$$

$u_b$ represents the unit vector pointing in the direction of the ball with respect to the character's COM, $v_{com}$ is the character's COM velocity, $\hat{v} = 1m/s$ is the target speed, $x_{com}$ is the COM position, and $x_b$ is the ball's position. Similarly, $r_t^{bv}$ and $r_t^{bp}$ encourages the character to move the ball towards the target,

$$r_t^{bv} = \exp\left(-1 \min\left(0, u_{tar} \cdot v_b - \hat{v}\right)^2\right)$$

$$r_t^{bp} = \exp\left(-0.5 \left\|x_b - x_{tar}\right\|^2\right).$$

$u_{tar}$ represents the unit vector pointing in the direction of the target with respect to the ball, $v_b$ is the velocity of the ball, and $x_{tar}$ is the target location. The weights for the reward terms are specified according to $(w^{cv}, w^{cp}, w^{bv}, w^{bp}) = (0.1, 0.1, 0.3, 0.5)$.

**Holdout:** The holdout task is based on the standard Gym `Ant-v3` environment. The goal $g_t = \left(\cos(\hat{\theta}), \sin(\hat{\theta})\right)$ specifies a two-dimensional vector that represents the target direction $\hat{\theta}$ that the character should travel in. The reward function is similar to that of the standard `Ant-v3` environment:

$$r_t = w^{\text{forward}} r_t^{\text{forward}} + w^{\text{healthy}} r_t^{\text{healthy}} + w^{\text{control}} r_t^{\text{control}} + w^{\text{contact}} r_t^{\text{contact}},$$

but the forward reward $r_t^{forward}$ is modified to reflect the target direction $\hat{u} = \left(\cos(\hat{\theta}), \sin(\hat{\theta})\right)$:

$$r_t^{\text{forward}} = \hat{u} \cdot v_{com}$$

where $v_{com}$ represents the character's COM velocity along the horizontal plane. The weights of the reward terms are specified according to $\left(w^{\text{forward}}, w^{\text{healthy}}, w^{\text{control}}, w^{\text{contact}}\right) = (1.0, 1.0, 0.5, 0.0005)$. During pre-training, the policies are trained with directions $\hat{\theta} \in [0, 3/2\pi]$. During transfer, the policies are trained with directions sampled from a holdout set $\hat{\theta} \in [3/2\pi, 2\pi]$.

## E  Model Setup

All models are trained using proximal policy optimization (PPO) [37], except for the option-critic model, which follows the update rules proposed by Bacon et al. [2]. A discount factor of $\gamma = 0.95$ is used during pre-training, and $\gamma = 0.99$ is used for the transfer tasks. The value functions for all models are trained using multi-step returns with TD($\lambda$) [40]. The advantages for policy gradient calculations are computed using the generalized advantage estimator GAE($\lambda$) [36]. We detail the hyperparmater settings for each model in the following sections.

### E.1  MCP

The MCP model follows the architecture detailed in Figure 3. The value function $V(s, g)$ is modeled with a fully-connected network with 1024 and 512 hidden units, followed by a linear output unit. Hyperparameter settings are available in Table 5.

| Parameter | Biped | Humanoid | T-Rex |
|---|---|---|---|
| $k$ Primitives | 8 | 8 | 8 |
| $\pi$ Stepsize (Pre-Train) | $2 \times 10^{-5}$ | $1 \times 10^{-5}$ | $1 \times 10^{-5}$ |
| $\pi$ Stepsize (Transfer) | $5 \times 10^{-5}$ | $5 \times 10^{-5}$ | $5 \times 10^{-5}$ |
| $V$ Stepsize | $1 \times 10^{-2}$ | $1 \times 10^{-2}$ | $1 \times 10^{-2}$ |
| Batch Size | 4096 | 4096 | 4096 |
| Minibatch Size | 256 | 256 | 256 |
| SGD Momentum | 0.9 | 0.9 | 0.9 |
| TD($\lambda$) | 0.95 | 0.95 | 0.95 |
| GAE($\lambda$) | 0.95 | 0.95 | 0.95 |
| PPO Clip Threshold | 0.02 | 0.02 | 0.02 |

Table 5: MCP model hyperparamters.

## E.2 Scratch

As a baseline, we train a model from scratch for each transfer task. The policy network consists of two fully-connected layers with 1024 and 512 ReLU units, followed by a linear output layer that outputs the mean of a Gaussian distribution $\mu(s, g)$. The covariance matrix is represented by a fixed diagonal matrix $\Sigma = \mathrm{diag}(\sigma_1, \sigma_2, ...)$ with manually specified values for $\sigma_i$. The value function follows a similar architecture, but with a single linear output unit. Hyperparameter settings are available in Table 6.

| Parameter | Biped | Humanoid | T-Rex |
|---|---|---|---|
| $\pi$ Stepsize | $2.5 \times 10^{-6}$ | $2.5 \times 10^{-6}$ | $1 \times 10^{-6}$ |
| $V$ Stepsize | $1 \times 10^{-2}$ | $1 \times 10^{-2}$ | $1 \times 10^{-2}$ |
| Batch Size | 4096 | 4096 | 4096 |
| Minibatch Size | 256 | 256 | 256 |
| SGD Momentum | 0.9 | 0.9 | 0.9 |
| TD($\lambda$) | 0.95 | 0.95 | 0.95 |
| GAE($\lambda$) | 0.95 | 0.95 | 0.95 |
| PPO Clip Threshold | 0.02 | 0.02 | 0.02 |

Table 6: Scratch model hyperparamters.

## E.3 Finetuning

The finetuning model is first pre-trained to imitate a reference motion, and then finetuned on the transfer tasks. The network architecture is identical to the scratch model. Pre-training is done using the motion imitation approach proposed by Peng et al. [32]. When transferring to tasks with additional goal inputs $g$ that are not present during training, the networks are augmented with additional inputs using the *input injection* method from Berseth et al. [3], which adds additional inputs to the network without modifying the initial behavior of the model. Hyperparameter settings are available in Table 7.

| Parameter | Biped | Humanoid | T-Rex |
|---|---|---|---|
| $\pi$ Stepsize | $2.5 \times 10^{-6}$ | $2.5 \times 10^{-6}$ | $1 \times 10^{-6}$ |
| $V$ Stepsize | $1 \times 10^{-2}$ | $1 \times 10^{-2}$ | $1 \times 10^{-2}$ |
| Batch Size | 4096 | 4096 | 4096 |
| Minibatch Size | 256 | 256 | 256 |
| SGD Momentum | 0.9 | 0.9 | 0.9 |
| TD($\lambda$) | 0.95 | 0.95 | 0.95 |
| GAE($\lambda$) | 0.95 | 0.95 | 0.95 |
| PPO Clip Threshold | 0.02 | 0.02 | 0.02 |

Table 7: Finetune model hyperparamters.

### E.4 Hierarchical

The hierarchical model consists of a gating function $w(s, g)$ that specifies the probability of activating a particular low-level primitive $\pi_i(a|s)$ from a discrete set of primitives. To enable the primitives to be transferable between tasks with different goal representations, the hierarchical model follows a similar asymmetric architecture, where the primitives have access only to the state. During pre-training, each primitive is trained to imitate a different reference motion. All experiments use the same set of 7 primitives, including 1 primitive trained to walk forwards, 3 primitives trained to turn right at different rates, and 3 primitives trained to turn left at different rates. Once the primitives have been trained, their parameters are kept fixed, while a gating function is trained to sequence the primitives for each transfer task. The gating function selects a new primitive every walk cycle, which has a duration of approximately 1 second, the equivalent of about 30 timesteps. Each primitive is modeled using a separate network with a similar network architecture as the scratch model. The gating function is modeled with two fully-connected layers consisting of 1024 and 512 ReLU units, followed by a softmax output layer that specifies the probability of activating each primitive. The gating function is also trained with PPO. Hyperparameter settings are available in Table 8.

| Parameter | Biped | Humanoid |
|---|---|---|
| $k$ Primitives | 7 | 7 |
| $\pi$ Stepsize | $1 \times 10^{-3}$ | $1 \times 10^{-3}$ |
| $V$ Stepsize | $1 \times 10^{-2}$ | $1 \times 10^{-2}$ |
| Batch Size | 4096 | 4096 |
| Minibatch Size | 256 | 256 |
| SGD Momentum | 0.9 | 0.9 |
| TD($\lambda$) | 0.95 | 0.95 |
| GAE($\lambda$) | 0.95 | 0.95 |
| PPO Clip Threshold | 0.02 | 0.02 |

Table 8: Hierarchical model hyperparamters.

### E.5 Option-Critic

The option-critic model adapts the original implementation from Bacon et al. [2] to continuous action spaces. During pre-training, the model is trained end-to-end with the motion imitation tasks. Unlike the hierarchical model, the options (i.e. primitives) are not assigned to a particular skills, and instead specialization is left to emerge automatically from the options framework. To enable transfer of options between different tasks, we also use an asymmetric architecture, where the intra-option policies $\pi_\omega(a|s)$ and termination functions $\beta_\omega(s)$ receive only the state as input. The policy over options $\pi_\Omega(\omega|s, g)$, as defined by the option value function $Q_\Omega(s, g, \omega)$, has access to both the state and goal. When transferring the options to new tasks, the parameters of $\pi_\omega$ and $\beta_\omega$ are kept fixed, and a new option value function $Q_\Omega$ is trained for the new task. We have also experimented with finetuning $\pi_\omega$ and $\beta_\omega$ on the transfer tasks, but did not observe noticeable performance improvements. Furthermore, joint finetuning often results in catastrophic, where the options degrade to producing highly unnatural behaviours. Therefore, all experiments will have $\pi_\omega$ and $\beta_\omega$ fixed when training on the transfer tasks. Hyperparameter settings are available in Table 9.

| Parameter | Biped | Humanoid | T-Rex |
|---|---|---|---|
| $k$ Options | 8 | 8 | 8 |
| $\pi$ Stepsize | $2.5 \times 10^{-6}$ | $2.5 \times 10^{-6}$ | $1 \times 10^{-6}$ |
| $\beta_\omega$ Stepsize | $2.5 \times 10^{-6}$ | $2.5 \times 10^{-6}$ | $1 \times 10^{-6}$ |
| $Q_\Omega$ Stepsize | $1 \times 10^{-2}$ | $1 \times 10^{-2}$ | $1 \times 10^{-2}$ |
| Batch Size | 256 | 256 | 256 |
| SGD Momentum | 0.9 | 0.9 | 0.9 |
| $\xi$ Termination Cost | 0.01 | 0.01 | 0.01 |

Table 9: Option-critic model hyperparamters.

### E.6 Mixture-of-Experts

The mixture-of-experts (MOE) model is implemented according to Equation 1. The policy consists of a set of primitives $\pi_i(a|s)$ and gating function $w(s, g)$ that specifies the probability of activating each primitive. To facilitate transfer, the primitives only receives the state as input, while the gating function receives both the state and the goal. The primitives are first pre-trained with the motion imitation task, and when transferring to new tasks, the parameters of the primitives are kept fixed, while a new gating function is trained for each transfer task. Therefore, MOE is analogous to MCP where only a single primitive is activated at each timestep. The gating function and the primitives are modeled by separate networks. The network for the gating function consists of 1024 and 512 ReLU units, followed by a softmax output layer that specifies $w_i(s, g)$ for each primitive. The primitives are modeled jointly by a single network consisting of 1024 and 512 ReLU units, followed separate linear output layers for each primitives that specifies the parameters of a Gaussian. As such, the MOE model's action distribution is modeled as a Gaussian mixture model. Hyperparameter settings are available in Table 10.

| Parameter | Biped | Humanoid | T-Rex |
|---|---|---|---|
| $k$ Primitives | 8 | 8 | 8 |
| $\pi$ Stepsize | $1 \times 10^{-5}$ | $5 \times 10^{-6}$ | $2 \times 10^{-6}$ |
| $V$ Stepsize | $1 \times 10^{-2}$ | $1 \times 10^{-2}$ | $1 \times 10^{-2}$ |
| Batch Size | 4096 | 4096 | 4096 |
| Minibatch Size | 256 | 256 | 256 |
| SGD Momentum | 0.9 | 0.9 | 0.9 |
| TD($\lambda$) | 0.95 | 0.95 | 0.95 |
| GAE($\lambda$) | 0.95 | 0.95 | 0.95 |
| PPO Clip Threshold | 0.02 | 0.02 | 0.02 |

Table 10: Mixture-of-experts model hyperparamters.

### E.7 Latent Space

The latent space model follows a similar architecture as Merel et al. [26], where an encoder $q(w_t|g_t)$ first maps the goal $g_t$ to a distribution over latent variables $w_t$. $w_t$ is then sampled from the latent distribution and provided to the policy as input $\pi(a_t|s_t, w_t)$. The latent distribution is modeled as an IID Gaussian $q(w_t|g_t) = \mathcal{N}(\mu_q(g_t), \Sigma_q(g_t))$ with mean $\mu_q(w_t)$ and diagonal covariance matrix $\Sigma_q(g_t)$. Similar to VAEs, we include a term in the objective that regularizes the latent distribution against a standard Gaussian prior $p_0(w_t) = \mathcal{N}(0, I)$,

$$\arg\max_{\pi,q} \quad \mathbb{E}_{\tau \sim p_{\pi,q}(\tau)} \left[ \sum_{t=0}^{T} \gamma^t r_t \right] + \beta \, \mathbb{E}_{g_t \sim p(g_t)} \left[ D_{KL} \left[ q(w_t|g_t) || p_0(w_t) \right] \right] \tag{12}$$

Here, $\beta$ is a manually specified coefficient for the KL regularizer. The encoder and policy are trained end-to-end using the reparameterization trick [20].

The latent space model follows a similar pre-training procedure as the MCP model, where the model is trained to imitate a corpus of reference motions with the goal $g_t = (\hat{s}_{t+1}, \hat{s}_{t+2})$ specifying the target states for the next two timesteps. The encoder is therefore trained to embed short motion clips into the latent space. After pre-training, the parameters of $\pi$ are frozen, and a new encoder $q'(w_t|s_t, g_t)$ is trained for each transfer task. Following the architectures from previous work [16, 26], the encoder used during pre-training only receives the goal $g_t$ as input, while the encoder used in the transfer tasks receives both the state $s_t$ and goal $g_t$ as input, since additional information from the state may be necessary when performing the new tasks.

The policy network follows a similar architecture as the ones used by the finetuning model, consisting of two hidden with 1024 and 512 ReLU units followed by a linear output layer. The encoder used during pre-training consists of 256 and 128 hidden units, followed by a linear output layer for $\mu_q(g_t)$ and $\Sigma_q(g_t)$. The size of the encoding is set to be 8D, the same dimensionality as the weights of the gating function from the MCP model. The encoder used in the transfer tasks is modeled by a larger network with 1024 and 512 hidden units. Hyperparameter settings are available in Table 11.

| Parameter | Biped | Humanoid | T-Rex |
|---|---|---|---|
| $w$ Latent Size | 8 | 8 | 8 |
| $\pi$ Stepsize (Pre-Train) | $5 \times 10^{-6}$ | $2.5 \times 10^{-6}$ | $1 \times 10^{-6}$ |
| $\pi$ Stepsize (Transfer) | $5 \times 10^{-5}$ | $5 \times 10^{-5}$ | $5 \times 10^{-5}$ |
| $V$ Stepsize | $1 \times 10^{-2}$ | $1 \times 10^{-2}$ | $1 \times 10^{-2}$ |
| Batch Size | 4096 | 4096 | 4096 |
| Minibatch Size | 256 | 256 | 256 |
| SGD Momentum | 0.9 | 0.9 | 0.9 |
| TD($\lambda$) | 0.95 | 0.95 | 0.95 |
| GAE($\lambda$) | 0.95 | 0.95 | 0.95 |
| PPO Clip Threshold | 0.02 | 0.02 | 0.02 |
| $\beta$ KL Regularizer | $1 \times 10^{-4}$ | $1 \times 10^{-4}$ | $1 \times 10^{-4}$ |

Table 11: Latent space model hyperparamters.