[Reviews · NeurIPS 2019]

Reviewer 1



The authors propose a method of combining multiple sub-policies with continuous action spaces by multiplicative composition (instead of the standard additive model in options,etc.). The sub policies are pre-trained with imitation learning. MCP shows competitive or much better results than other state of the art hierarchical and latent space methods on challenging high-dimensional domains (the T-Rex playing soccer!). Pros: 1) The idea is clearly written and with several details for re-implementation 2) Compelling results on challenging environments 3) Good baseline comparisons with very recent papers 4) The analysis with latent space methods is really appreciated Cons: 1) I’m not sure how novel the idea is, as there is a lot of literature on using an ensemble or mixture of experts/dynamics models/policies, etc. That being said, the results are very compelling. A possible area of overlap is with the motion generation literature which the paper does not discuss e.g. Riemannian Motion Policies/RMPFlow (Ratliff et.al) are used to combine multiple policies in a modular and hierarchical fashion and transfer between different spaces. Discovery of Deep Continuous Options by Krishnan et al. may also be relevant to discuss. 2) I’m not sure if the option-critic baseline is fair. As far as I understand, OC learns the options (they don’t need to be hand defined) and in MCP there is a clear separation between pre-training the skills and learning the gating. Perhaps a better baseline would be to pre-train the options in OC (in the same way as MCP) and compare against the full MCP pipeline. Then, we can see 3) Do you hand-define the training for the sub policies (its stated that there are 8 sub-skills). For instance, skill #1 is for walking, #2 is using the foot to kick the ball, #3 is for turning, etc.? 4) What are the crucial hyperparameters in MCP? Some insight regarding this would be useful. 5) Since the paper claims to beat recent state-of-the-art methods (e.g. Figure 4) in non-standard environments like the T-Rex dribbling (i.e. its not an open ai gym task), the authors should release code.

Reviewer 2



Originality: I’m not very familiar with RL and imitation learning, but the work seems original. The word directly addresses a deficiency in existing approaches to learning complex behaviors like mixture-of-experts and hierarchical models. Quality: The quality of the work seems high overall. The explanation of the model is fairly clear, the experiments seem thorough, and there is abundant followup work suggesting that the model is achieving the desired effect, namely that the model is learning a set of primitive experts that combine to learn complex behaviors (Figure 7). Clarity - I found the terminology around a primitive’s “activation” to be quite confusing. My understanding is that a primitive being “active” means that it is contributing to the distribution that is actually sampled from. Under this definition, it makes sense that we would want multiple primitives to be active so that we can leverage the representational power of K many models rather than just one, in addition to primitive specialization. On the other hand, when you mention activating “primitive skill”, you seem to suggest performing multiple actions at the same time step. Does this mean that that model is allowed to activate several actions at each time step? That doesn’t seem to be the case in the studied setting, but seems to be used to justify MCP. - Using the definition of a primitive being active meaning they contribute to the sampling distribution, in the additive model, it seems fairly trivial to sample in a way such that multiple primitives could be “active”: Instead of sampling from w, compute the linear combination of the primitives and sample from the resulting distribution. If that’s the case, this seems like a useful baseline in the experiments for isolating the effect of the representational power of the particular model used (multiplicative factoring primitives) versus the ability to have multiple primitives active. Significance: Overall the work seems quite significant. The theoretical benefits of the proposed model allows the model to learn complex behaviors, which seem to indeed play out in experiments. I could see work building directly on the model presented here, and comparing to it as a baseline.

Reviewer 3



Generally speaking,this is a very interesting work with great originality and quality. However, the formulas and principles are not very clear. For example, how does the Gaussian primitives generate another Gaussian, and must the Gaussian policies be used? What is even more puzzling is that the addition model can't really be able to activate multiple primitives simultaneously as the multiplication model? From the perspective of the formula, the two are just the difference between addition and multiplication. Finally, I believe that learning and composing skills effectively are very meaningful research directions, and this paper does a significant job.

[Author Response · NeurIPS 2019]

We would like to thank the reviewers for their insight and suggestions. We will aim to incorporate your feedback in future drafts of the paper.

**Re: primitive specializations**

We would like to clarify that we do not manually assign skills to each primitive. As mentioned at the end of section 3.1 and the start of 3.2, the primitives are trained jointly, and the specializations emerge automatically from the learning process. For example, we do not assign one primitive for walking, and then another for turning. Instead the primitives learn to specialize in different low-level skills that can then be composed to produce different walks, turns, or kicks. We will adjust our writing to improve the clarity on this point.

**Re: code release**

If accepted, we intend to release the code for MCP, the environments, and the motion dataset used for pre-training.

**Individual Responses:**

**Reviewer 3:**
**Re: Gaussian primitives**

The formulation of MCP in Equation 2 is not restricted to Gaussian primitives. Other distributions can also be used, as long as the product of the primitives produce a tractable distribution. We chose to use Gaussian primitives in this work, because they are widely used for continuous control tasks, and provide a simple analytic form for the composite distribution, since the product of Guassians is another Gaussian. We will include a proof of this property.

**Re: activating multiple primitives with the additive model**

Thank you for raising this point, we will improve the description of our method to better clarify this. The crucial difference that enables MCP to activate multiple primitives simultaneously is indeed the multiplicative composition scheme. Adding the densities of the primitive distributions produces a mixture policy, where the agent selects a single primitive per time step. Therefore, the action at any given time step would come from only one of the primitives. Multiplying the primitives together instead fuses their distributions, intuitively producing a new distribution that is their "intersection". If one were to modify the additive model to activate multiple primitives by adding the samples from the individual distributions, then this will no longer result in the additive composite distribution described in Equation 1. In fact, this will produce a composite distribution that is more akin to the MCP model. We will add this discussion to the paper to better clarify the distinction.

**Reviewer 1:**
**Re: option-critic baseline**

The options in the option-critic baseline are also pre-trained in the same manner as MCP, using the motion imitation tasks, and the learned options are then transferred to new tasks. We have also experimented with training an option-critic model from scratch on the transfer tasks, but found that pre-training yielded better performance.

**Re: additional references and ensemble/mixture policies**

Thank you for the pointers, we will include these additional references. In principle, many of the algorithms proposed in prior work for learning options and mixture policies, including DDCO [Krishnan et al., 2017], could also be used to train multiplicative primitives, as in our work – in that sense, our contribution is complementary to these prior methods and largely orthogonal. However, we also observe that the relatively simple training procedure in our work is effective at learning useful multiplicative primitives on a range of difficult tasks.

**Re: hyperparameters**

Tables with the hyperparameters used for MCP and other baselines are available in the supplementary material. We will also include more detailed explanations for the different parameters in the final draft.

**Reviewer 2:**
**Re: activating multiple actions per timestep**

We will improve the writing to better clarify the use of "activation". The compose policy still proposes a single action per timestep, but the distribution from which the actions are sampled from is the result of composing multiple primitive distributions, which may each specialize in different skills. For example, as shown in the supplementary video and Figure 7, we observe that some primitives specialize in pushing back with a particular leg, while other primitives specialize in lifting the legs. By composing these primitives, MCP can then produce a range of different behaviors.

**Re: ablation experiments**

Thank you for the suggestions. We will include additional ablation experiments for fixing the primitive weights during transfer, and hiding the goal input from the primitives. One of the primary motivation for not providing the primitives with the goal is to prevent degeneracy. During pre-training, if the primitives also have access to the goal, then it is possible for a single primitive to solve all tasks, while the other primitives become inactive. This degeneracy can lead to poor transfer performance, as the other primitives may not acquire useful skills.

[Meta-Review · NeurIPS 2019]

This paper proposes an interesting method that learns multiple composable policies that can produce a continuous spectrum of diverse actions, where primitives can be transferred to new tasks. The paper is well written, and the experimental results are convincing. The ablation study would strengthen the paper.